# Establishment and Application of an Evaluation Model for Orchid Island Sustainable Tourism Development

**DOI:** 10.3390/ijerph16050755

**Published:** 2019-03-02

**Authors:** Han-Shen Chen

**Affiliations:** 1Department of Health Diet and Industry Management, Chung Shan Medical University, No. 110, Sec. 1, Jianguo N. Rd., Taichung City 40201, Taiwan; allen975@csmu.edu.tw; Tel.: +886-4-2473-0022 (ext. 12225); 2Department of Medical Management, Chung Shan Medical University Hospital, No. 110, Sec. 1, Jianguo N. Rd., Taichung City 40201, Taiwan

**Keywords:** sustainable tourism, island destination, environmental impact, recreation economics

## Abstract

Global warming and climate change increase the likelihood of weather-related natural disasters that threaten ecosystems and consequently affect the tourism industry which thrives on the natural attributes of island regions. Orchid Island, the study area, is home to the Yami (Tao) tribe—the only indigenous people of Taiwan with a marine culture. The island possesses rich geological and topographical features (such as coral reefs) and distinctive biological and ecological resources (such as the green sea turtle, flying fish, and Orchid Island scops owl), and organizes traditional festivals and activities (such as the flying fish festival) as well as tribal tourism activities. These factors contribute to its immense potential to become the new tourism hotspot. To study the factors enhancing tourist experiences, a random utility model was constructed using a choice experiment method (CEM) for the tourist resort on Orchid Island. The study results demonstrated that: (1) Limiting tourists to 600/day; employing professional tour guides; providing better recreational facilities; introducing additional experience-enhancing activities; and lowering contributions towards the professional ecosystem conservation trust fund will improve the overall effectiveness of attracting tourists to Orchid Island. The evaluation results from both conditional logit and random parameter logit models were similar; (2) the analysis results from the latent class model demonstrated that island tourism has significant market segmentation. The socioeconomic backgrounds of tourists, their experiences, and their preferences exhibit heterogeneity, with significant differences in willingness to pay for island tourism.

## 1. Introduction

According to the “Fourth Assessment Report (AR4) of the United Nations Intergovernmental Panel on Climate Change (IPCC),” global warming and climate change have become increasingly serious. The resulting rise in sea level from climate change can negatively impact water resources, ecological balance, and the environment. Climate change has a more severe impact on islands that are surrounded by the sea. Relevant studies have pointed out that besides threatening ecosystems, natural disasters will also affect the operations of the tourism industry [1,2]. The UN World Tourism Organization (2011) mentioned that extreme weather conditions may affect 1.8 billion international tourists in 2030. The direct effects of climate change on the tourism industry include losses to manpower and property. This also leads to a significant decrease in tourism as more and more tourists decide not to travel to such sites. Climate change will also indirectly affect the environment and culture of the tourist destination, the economic output of the tourism industry, and the damage infrastructure [3]. The increasing demands for recreational tourism, changing tourist preferences, increasing awareness on ecological conservation, and the rapid development in island tourism in recent years has attracted international researchers interested in island development to conduct relevant studies [4,5,6].

Dahlin et al. [7] pointed out that the development of island tourism will inevitably lead to some signs of imbalance, including excessive coastal development, the destruction of ecological resources, pollution created by waste, etc. It may also lead to land encroachments to meet increasing demands for accommodation and recreational water activities for tourists. Such changes may affect the traditional community system and gradually create an imbalance in the lifestyles of the locals. Therefore, while islands may provide sightseeing and recreational services, they might simultaneously experience the negative effects caused by the development in tourism on the environment, society, and local culture due to their limited area and natural resources, vulnerability to disasters, compromised ability to recover from disasters, and economic dependence [8,9,10].

However, the development of the tourism industry is accompanied by increasingly prominent environmental problems. As the ultimate goal of sustainable tourism is still the sustainable management of tourism destinations, the main application methods are still based on the environmental carrying capacity, land division management, visitor impact management, and sustainable development indicators. Zhang et al. [11] suggested that, although existing sustainability evaluation methods can be used to assess the effects of human activities on ecosystem functions, they have a limited application in evaluating social economy. The use of this type of analysis to evaluate the economic benefits of tourism in that area can aid the concerned management in making related decisions on planning and utilization and/or sustainable operation of local ecologic resources. In previous assessments of the benefits of recreational resources, many studies have employed the travel cost method and contingent valuation method (CVM) to assess the benefits of island tourism [12,13,14]. However, as travel cost methods and CVM methodologies have certain limitations in application, the CEM has gradually become a significant assessment method for conducting preference studies on the conservation of natural resources in recent years. CEM is also an important assessment tool for measuring the value of nonmarket goods [15].

Liekens et al. [16] pointed out that besides the simultaneous assessment of use and nonuse values, CEM can also define the hypothetical market through questionnaire surveys to understand public preferences for landscape conservation and natural development. This will further reflect the value of environmental goods (or services). As CEM has a multiattribute and multilevel assessment ability, different combinations of alternative programs are used to assess the important characteristics of nonmarket goods or services. Choice sets of different hypothetical scenarios are used to enable the respondents to select appropriate alternative programs based on their preferences. This avoids an assessment bias [17]. Due to the aforementioned advantages, CEM has also been used to evaluate factors with nonmarket value in recent years, including species conservation [15,18,19,20]; wetland rehabilitation [21,22,23]; island tourism preferences [24,25,26,27,28]; and coastal region conservation [29,30,31,32,33,34]. In addition, CEM was also employed to examine tourist preferences for land and environmental functions in national parks [35,36]. Other studies that employed CEM focused on how to change specific ecosystem services to affect economic benefits [37,38,39,40].

For empirical models, conditional logit (CL) models can be used to estimate the average preference of tourists from the multiple attributes of island tourism and to estimate the marginal willingness to pay (MWTP) for these attributes [28,33]. The random parameter logit (RPL) model can reflect the different responses of respondents from different backgrounds towards different attributes. This can be used to examine the heterogeneous preferences of respondents and their willingness to pay for changes in the levels of various attributes (such as folk and cultural experiences, ecologic experiences, and other attributes) [41,42,43,44].

To segment a clearer target market, the latent class model (LCM) can segregate respondents into different groups and examine and compare their preferences and group differences (such as the island tourism preferences, attitudes, and socioeconomic background of interviewed tourists) [26]. From the aforementioned studies, we can see that the empirical CEM models of CL, RPL, and LCM have been verified for use in the examination and evaluation of multiple attribute preferences at island tourism sites.

Previous studies have shown that prior economic evaluation of nonmarket resources has mostly focused on the evaluation of forests, coastal areas, natural parks, and nature reserves. Only a few studies have evaluated the recreational value of island tourism. Therefore, this study used CEM to construct an island tourism attribute utility model and further employed CL and RPL models to estimate the utility function of island tourism. The socioeconomic background, awareness, and tourists’ attitudes towards island tourism were considered to examine the differences in MWTP for various attributes. LCM was used to test whether respondents had heterogeneous preferences for island tourism so that they could select appropriate alternative programs based on their preferences. These will be used to evaluate the ecological environment, socioeconomic situation, and tribal tourism programs on Orchid Island to determine their economic benefits and to create a system of sustainable development. The aforementioned methodology has representativeness and research originality and can compensate for the current lack of studies on sustainable tourism development on islands in academia. The research contribution made by studying the aforementioned problems can assist the world, particularly the academic world, by providing a reference for economic evaluation models for sustainable island tourism and operation and management strategies.

The aim of this study is to evaluate the economic benefits of tourism for Orchid Island. A CEM is used to estimate the average preferences of tourists from the multiple attributes of island tourism and to estimate the WTP for these attributes. The study is divided into four parts. In Section 1, the motivation for the study is discussed, and the study objective is proposed. In Section 2, we construct the preference utility model for island tourism and introduce preferred selection combinations for choice sets for the Orchid Island tourism site. In Section 3, we analyze the results of factors influencing multiple attribute preferences, along with their WTP. In Section 4, based on the results, countermeasures and suggestions are proposed for the sustainable development of the Orchid Island environment, providing a reference for policy makers to make more efficacious policies.

## 2. Literature Review

The CEM is a stated preference evaluation technique. Respondents are given multiple choices and forced to make trade-offs between them. Each option is described in terms of a bundle of attributes describing the good presented at various levels. The principle advantage of CEM is the ability to value individual characteristics of environmental goods and the marginal value of changes in characteristics.

In the past, economists try to assess people’s WTP for ecotourism preferences [24,25,33], species conservation [19,20], and, more recently, also the issue of WTP for ecosystem services was explored by many authors [37,38,39,45] but less frequent for evaluation of island tourism preferences [27,28]. Remoundou et al. [31] employed CEM to evaluate the effects of climate change on the willingness to pay for Santander’s coastal ecosystem. The study attributes included biodiversity, jellyfish blooms, days when the beaches were closed, sizes of the beaches, and annual additional household expenditure. Viteri Mejía and Brandt [33] employed CEM to interview tourists visiting the Galapagos Islands to assess their willingness to pay for protective measures against invasive species. The study attributes included depth of experience in the islands’ ecosystem, length of stay, level of protective measures taken against invasive species, and price of island tourism. The results of that study showed that tourists visiting the Galapagos Islands highly valued the biodiversity on the island and were marginally willing to pay USD $2543 for better protective measures. Schuhmann et al. [32] employed CEM to evaluate tourist preferences and willingness to pay for coastal attributes in Barbados. The study attributes included price, type of accommodation, beach width, distance from beach, and beach litter.

Cazabon-Mannette et al. [29] employed CEM and CVM to evaluate the nonuse value and nonconsumptive value of sea turtles in Tobago. The study attributes included price, number of sea turtle sightings, fish diversity (number of species), coral cover, and degree of congestion (number of divers). Xuan et al. [34] used a discrete choice experiment to evaluate tourists’ willingness to pay for boat tours in the marine protected area of Vietnam’s Nha Trang Bay. The study attributes included coral cover, environment quality, rate of unemployment of fishermen, and increase in ticket prices. Peng and Oleson [30] employed a discrete choice experiment to evaluate beach recreationalists’ preferences and willingness to pay to improve the water quality of Oahu beaches. The study attributes included water quality, water turbidity, coral cover, fish diversity, and willingness to pay for motor vehicles. Park and Song [41] applied a latent profile analysis (LPA) and CEM to identify latent classes based on visitors’ perceived place value and to estimate visitors’ willingness to pay (WTP) in an Urban Lake Park.

The above studies showed varying levels of WTP depended on factors such as where the study was conducted, products and product attributes included and, data collection and analysis methods used. In fact, through the questionnaire, the socioeconomic background (such as age, education, marital status, and income) of respondents and respondents awareness and behavior (such as environmental attitude, perceived value, and revisit intention) towards island tourism were used as perspectives to examine the differences in WTP for various attributes. Halkos and Matsiori [46] pointed out that the comparative study of residents’ and tourists’ WTP for improving the quality (protection) of the Pagasitikos coastal area in Greece found that income, education, environmental attitude are the most important factors affecting payment amount. Tonin [47] indicated, the previous knowledge or familiarity with coralligenous habitats and biodiversity issues, income, education, environmental attitudes are main positive and significant determinants of WTP. The purpose of this study is the application of the CVM to the benefits of Orchid Island tourism management programs and the quantification of their value. In this case, it is not only the value of the island tourism that is calculated, but rather the economic benefits as a whole is evaluated through the respondents’ opinion of the goods and services produced.

## 3. Materials and Methods

### 3.1. Description of the Study Area

Taiwan is surrounded by a coastline of 1141 km and has abundant marine resources. Island tourism in Taiwan has natural and ecological features as well as historical and cultural features and is an emerging tourist destination [48]. The study site, Orchid Island, which has a total area of 48 km^2^ and 5069 residents, is located southeast of Taiwan and is surrounded by the sea (Figure 1). It is home to Taiwan’s only group of indigenous people with a marine culture—the Yami (Tao) tribe—and has rich geological and topographical features (coastal terrain, coral reefs); distinctive biological and ecological resources (green sea turtle, flying fish, coconut crab, Orchid Island scops owl, and Arius (Podocarpus costalis)); traditional festivals and activities (launching ceremony, flying fish festival); and aboriginal settlements. Orchid Island has witnessed a gradual development in diverse theme-based tours, which include natural ecology-based tours and relevant experience activities (snorkeling, whale watching, night observation of flying fish, and so forth). In recent years, supported by government policies, island tourism and tribal tourism have become the new tourism trends in Taiwan and have significant developmental potential for the future.

However, the growth of the tourism industry is accompanied by a detrimental impact on the environment. The construction of coastal embankments and tetrapods causes severe damage to the coastal environment and destroys biological habitats. In addition, tourism also indirectly affects the unique tatala boat culture of the Yami (Tao) people. The invasion of foreign culture causes a heritage crisis in the traditional, cultural, and social structure of the Yami (Tao) people. As island ecosystems are fragile, island development should emphasize the development of state land for environmental and cultural conservation and protection while developing unique ecological and cultural experiences to promote its tourism. The development of sightseeing resources must consider sustainable ecological, economic, and social development and try to minimize the impact of recreational activities on the environment. Therefore, development of the island tourism industry based on sustainable operating principles and conservation of the environment and its ecosystem are important topics to consider for the sustainable development of island tourism.

### 3.2. Construction of Preference Utility Model for Island Tourism

#### 3.2.1. Multi-Attribute WTP Valuation Model

First, CEM was used to construct an island tourism attribute utility model. Following that, CL and RPL models were used to estimate the utility function of island tourism. The socioeconomic backgrounds of tourists and tourist awareness and behavior towards island tourism were used to examine the differences in MWTP for various attributes. LCM was used to test whether there were heterogeneous preferences for island tourism present in respondents. Lastly, the aforementioned empirical analysis results were used to estimate the economic benefits of island tourism.

CEM is a standard random utility model (RUM). Therefore, it was used to explore the MWTP for all attributes and levels [49]. In the binary model, the utility of the n^th^ respondent is assumed to be the various options it faces ($U_{\mathrm{ni}}$), and the options are used to maximize the utility, as shown in Equation (1):(1)$$U_{\mathrm{ni}}=V_{\mathrm{ni}}+\varepsilon_{\mathrm{ni}}$$
where $U_{\mathrm{ni}}$ represents the attribute of the n^th^ respondent facing the i^th^ option, $V_{\mathrm{ni}}$ represents the observable part of the utility function, and $\varepsilon_{\mathrm{ni}}$ represents the residual item, i.e., the unobservable part.

This study intended to explore differences in preferences and WTP between respondents of different social and economic backgrounds, considering various attributes and levels. The analysis was conducted using a random parameters logit (RPL) model. The overall utility of the RPL model was determined as follows:(2)$$U_{\mathrm{ni}}=V_{\mathrm{ni}}\left(X_{\mathrm{ni}},S_{n}\right)+I{\_}_{\mathrm{ni}}$$
where $V_{\mathrm{ni}}$ is the utility coefficient of observable variable $X_{\mathrm{ni}}$ and respondent characteristic $S_{n}$ and represents the respondent’s preference, and $\varepsilon_{\mathrm{ni}}$ is the residual item.

To estimate the relative importance of all attributes of the product in terms of value, it was assumed that the degrees of various attributes in the alternative plan remained the same. Then, the marginal change in WTP for the kth attribute was determined by Equation (3):(3)$$\mathrm{WTP}=\frac{-I2_{k}}{I2_{c}}$$
where ${I2}_{k}$ is the attribute *k* parameter and ${I2}_{c}$ is the payment tool parameter.

#### 3.2.2. Introduction to Multiple Attributes and Levels of Orchid Island Tourism

Hanley et al. [50] pointed out that after defining the evaluation attributes to be included in CE, the evaluation of the levels of the attributes is a relatively important process. They also pointed out that these levels should be specific and feasible for future application, and they can be formulated through literature reviews and expert interviews. Therefore, besides conducting reviews of relevant literature, this study also conducted interviews with five academics who were experts on the subject, two tour guides of the Orchid Island aboriginal tribe, and relevant staff from the public sector. Following that, four attributes, “limit on tourist numbers,” “tour guide system,” “recreational facilities,” and “experience activities”, were set up. In addition, an “ecosystem conservation trust fund” that represents currency variables, was used as an expenditure tool attribute. We further delineated the current status of various attributes (Table 1) for use as the basis for measuring changes in the levels of attributes. After expert interviews, we obtained the following recommendations for level settings:Tourist numbers should be controlled with the current level of 1000 tourists per day as the upper limit. Further discussions with experts resulted in a recommendation of limiting numbers to no more than 600 tourists per day as a principle;Professional tour guides should be provided to offer guided tours;Recreational facilities with minimal environmental impact should be planned;Activities that enhance the experience of local characteristics/culture, such as “ecotourism,” “tribal ceremonies,” and “cave and underground dwellings”, should be included in the scope of experience activities; andFindings from the expert interviews should be used to set the evaluation levels for the ecosystem conservation trust fund. This study defined the various attributes and their levels for Orchid Island tourism, as shown in Table 1 below:

### 3.3. Introduction to Preference Selection Combinations for Choice Sets for the Orchid Island Tourism Site

In order to understand the choices of tourists regarding multiple attribute preference programs for the Orchid Island site, a more precise improvement plan and the preference for each attribute level will need to be more clearly defined. These attributes included “limit on tourist numbers,” “tour guide system,” “recreational facilities,” “experience activities,” and “ecosystem conservation trust fund.” Further information on these attributes are introduced and examined in Table 1 (attributes and levels of attributes of the Orchid Island tourism site). The arrangement combinations of various attributes and their levels produced 288 possible factor combinations (3 × 2 × 2 × 4 × 6 = 288).

In actual operations, every respondent had to fill in their answers, i.e., select one of the three choice sets (the two alternative programs and one status quo alternative). If the respondent was unable to make a decision, they could select “uncertain”, and this choice set was considered as a missing value. The researcher explained the various attributes of the Orchid Island tourism site and their levels (Table 2) and the content of the choice sets for Orchid Island preferences to each tourist respondent. This was in order to make the tourist respondents understand the content of the preference attributes of the Orchid Island tourism site before they selected their preferences. In terms of questionnaire content presentation, the first part of the questionnaire, which was divided into “Orchid Island tourism development awareness and behaviour”, and the third part, which included “basic personal information”, were identical in all five versions of the questionnaire.

### 3.4. Survey Design

This study employed purposive sampling and one-to-one interviews with 438 tourists in Orchid Island, and, after factoring out the invalid questionnaires (i.e., those with omitted answers, incomplete answers, or those in which answers to all the questions received the same scale points were all deemed as invalid and removed), a total of 385 valid ones were collected, giving a recovery rate of 87.9%. With regard to the socioeconomic backgrounds of tourists, there were more males (*n* = 202)—52.5% of the total sample. In terms of age distribution, most people fell into the age groups of 31–40 years (*n* = 162%, 42.1%) and 20–30 years (*n* = 129%, 33.5%). In terms of education level, tourists with university education (*n* = 162%, 42.1%) made up the bulk of tourists. In terms of average personal monthly income, most tourists had an income of TWD 30,000–40,000 (inclusive) (*n* = 169%, 43.9%), followed by TWD 20,000–30,000 (inclusive) (*n* = 101%, 26.2%). Recreational activities that tourists engaged in (multiple selection allowed) were mostly water activities (62.4%), tribal ceremonies (52.7%), and tasting of the local cuisine (28.7%). Expenditure on Orchid Island (including participating in activities, buying souvenirs, etc.) was TWD 5000–10,000 (inclusive) (43.2%), followed by TWD 10,000–15,000 (inclusive) (32.7%). When asked whether they would agree to pay a sum towards the ecosystem conservation trust fund to support sustainable tourism development, most respondents were agreeable (*n* = 316%, 82.1%).

## 4. Results

### 4.1. Results of the Analysis of Factors Influencing Multiple Attribute Preferences for the Orchid Island Tourism Site

This study first used CL and RPL to estimate the utility functions of the multiple attributes of the Orchid Island tourism site and obtained the relevant factors that affected the functions. Table 3 shows the empirical results. With the significance levels of the factors ranging from 1% to 9%, the evaluation model of this study passed the goodness of fit test (likelihood value was 614.6, which was significantly greater than the threshold value of 21.7). From this, we can understand that the utility functions of multiple attributes of island tourism have good explanatory power [24,26,51]. The following section will further describe the empirical analysis results of both models.

At a 5% significance level, the coefficient of LIM^−−^ was positive and significant. From this, we understand that decreasing the current daily tourist limit from 1000 to 600 could increase the utility value of island tourism for tourists. At a 1% significance level, GUI^+^ and REC^+^ t-values were also very significant. From this, we deduce that formulating and implementing a system of tour guides and adding recreational facilities and improving the quality of the existing facilities will increase tourists’ preference for the Orchid Island tourism site. With regard to the ecosystem conservation trust fund, the t-value st a 1% significance level was negative and significant. From this, we can observe that the utility value of Orchid Island tourism will decrease for tourists by increasing the ecosystem conservation trust fund. RPL estimation results along with CL showed that only EXP^+++^ estimation results were positive and achieved a significance level of 10%. From this, we understand that adding three tourist activities to Orchid Island could significantly increase the utility levels for tourists.

### 4.2. Examination and Analysis of Benefits of Island Tourism Management Programs

To analyze the benefits of island tourism management programs, this study used CL attribute level parameters to estimate the WTP for various levels of attributes (as shown in Table 4). In Table 4, the WTP was calculated based on attribute coefficients in the CL model to represent the mean WTP of all respondents. From the empirical analysis results, we can see that under the optimal program, each tourist could generate TWD 1202 for every visit they participate in. Therefore, there will be a loss of value if the status quo is maintained. The level of attributes in the optimal program could increase benefits, i.e., an increase of TWD 2337 in WTP would occur with the optimal program compared with the status quo. Therefore, if the daily tourist limit can be decreased from the current number of 1000 tourists to 600 tourists (LIM^−−^), the professional tour guide system could be implemented (GUI^+^), recreational facilities could be added, the facility quality of island tourism sites could be improved (REC^+^), and the experience activities available could be increased from snorkeling, whale watching, and night observation of flying fish to include ecotourism, tribal ceremonies, and cave and underground dwelling experiences (EXP^+++^). This would be the most effective management program for improving the economic value of Orchid Island tourism. Thus, improving various attribute levels should prove to be a more efficient management program as compared with the one being currently implemented.

### 4.3. Examination of Willingness to Pay and Market Segmentation

The following section will further compare the socioeconomic backgrounds and behaviors of tourists in terms of their willingness to pay for the aforementioned levels of attributes. From Table 5, we can see that at a 10% significance level, the willingness to pay value for LIM is significantly different among individuals of different educational levels and is associated with intention to pay for the ecosystem conservation trust fund. In addition, tourists with tertiary education and above and tourists who are willing to pay for the ecosystem conservation trust fund have a higher willingness to pay to bring down the daily tourist number from 1000 to 800. The willingness to pay value for GUI^+^ was significantly different between sexes and age groups. Females and respondents above 30 years old indicated a higher willingness to pay for the implementation of the tour guide system. When examined for improving environmental and facility quality (REC^+^) at the 1% significance level, significant differences in tourist spending were identified. We can see that tourists who spend more have a higher willingness to pay for improving the quality of recreational facilities at the Orchid Island tourism site. Finally, at the 1% significance level, the willingness to pay for EXP^+++^ was associated with significant differences in intention to pay for the ecosystem conservation trust fund. This shows that tourists who are willing to pay for the ecosystem conservation trust fund have higher willingness to pay for three additional tourism activities. Previous studies utilizing RPL to examine the heterogeneous preferences of tourists and market segmentation showed similar results to this study [24,26,51].

Lastly, this study utilized LCM to construct a market segmentation model for Orchid Island tourism based on the aforementioned background differences in order to examine the differences in island tourism preferences and willingness to pay between different tourism groups. From Table 6, we can see that two potential market segmentation groups showed differences in island tourism preferences. The first group of tourists showed preferences for “reducing the daily tourist limit to 600 tourists”, “improving environment and facility quality”, and “increasing experience activities to three items”, and had lower preference for “ecosystem conservation trust fund”. In contrast, the option “increasing two experience activities” had a lower preference. The second group only showed significant differences in the utility function for “improving environment and facility quality” and “ecosystem conservation trust fund”, and their WTP for “improving environment and facility quality” was lower than that of the first group. The first categorical model included improvement in environment quality, and other attribute parameter preferences were relatively obvious. They can be classified as tourists with obvious preferences (accounting for 79.5% of the sample). In comparison, the second group was only focused on the environment quality and can be classified as tourists with a single preference (accounting for 20.5% of the sample). A comparison of the socioeconomic backgrounds and tourism behaviors of these two groups showed that tourists with obvious preferences are mostly females, have higher education levels, higher spending capacities, and are willing to pay for the ecosystem conservation trust fund. This group obtains relatively higher island tourism benefits within a specific attribute combination. LCM was used for market segmentation of tourists to Orchid Island to allow targeted market sales and self-positioning.

## 5. Discussion

Regarding the empirical results, CL and RPL produced similar evaluation results for Orchid Island tourists, and the tourists were shown to prefer a change in the status quo. In addition, RPL also reflected a heterogeneous distribution pattern for tourists’ preferences for the various attribute parameters. The only preference that showed an identical effect on tourists’ choices was “improving recreation quality.” The two models showed that tourists’ preferences included having a system of professional tour guides, improving the recreation and facility quality, adding three experience activities and decreasing the charges towards the ecosystem conservation trust fund. The results of previous studies on Costa Rica tourists’ preferences for ecotourism [52] and preferences of tourists for the Barva Volcano Area in Costa Rica [51] showed that tourists prefer improvements in infrastructure. These results are consistent with the results of this study that showed preferences for improved recreation and facility quality.

In addition, tourists indicated that they would prefer to simultaneously experience three activities—ecotourism, tribal ceremonies, and cave and underground dwellings—on Orchid Island to experience the natural, cultural, and ecologic landscapes. Tourists who were interviewed preferred the most diverse ecotourism experience program. The results of the study by Chaminuka et al. [25] on tourists’ preferences for ecotourism in the villages adjacent to the Kruger National Park in South Africa also support the results of this study. They found that tourists have relatively high MWTP for visits to villages and craft markets in these villages.

This study found that the socioeconomic background and tourism behavior of different tourists were associated with significant differences in the willingness to pay value for various attributes. Tourists with tertiary education and above who were willing to pay for the ecosystem conservation trust fund indicated a greater preference for decreasing the daily limit of tourists than tourists with an education level lower than tertiary education and who were not willing to pay for the ecosystem conservation trust fund. The former showed a greater willingness to pay to restrict the number of tourists from 800 to 600. Tourists who are willing to pay for the ecosystem conservation trust fund were more willing to pay value for the three experience activities than tourists who were not willing to pay for the ecosystem conservation trust fund. Tourists aged above 30 and females indicated preference for an explanatory tour guide system, while tourists who spend more were relatively more willing to pay for the improvement of recreation facilities. Previous studies have pointed out that there are differences in environmental attitude between residents and tourists, which vary according to gender, age, educational level, and other variables have similar results in this study [46,47]. 

The results of the analysis of preferred programs for developing island tourism in the areas of Orchid Island natural, cultural, and ecological landscapes showed that, under the current program, the benefit for each tourist’s visit is TWD 1135. If the program with the highest value was used for estimation, each tourist could provide a benefit of TWD 1202 for every visit. Therefore, maintaining the status quo will decrease the economic value of island tourism development. In contrast, the tourists’ willingness to pay value under the optimal program was increased by TWD 2337 compared with under the current program. This optimal program involves decreasing the daily limit of tourists from 1000 to 600 (TWD 269), implementing a system of professional tour guides (TWD 214), improving the environment and facility quality of the island tourism sites (TWD 518), and expanding the tourism experience activities available on Orchid Island from snorkeling, whale watching, and night observation of flying fish to include ecotourism, tribal ceremonies, and cave and underground dwelling experiences (TWD 201). This is the best management program for developing island tourism for Orchid Island. Lastly, this study used LCM to examine the market segmentation and heterogeneity in tourists’ preferences for Orchid Island tourism and classified tourists into tourists with obvious preferences and tourists with single preferences. The former accounted for 79.5% of the respondents and these individuals showed higher willingness to pay than the latter under the optimal program. This group of people are the market segmentation subjects that operators should focus on. This segment consists mostly of females who have a tertiary level education and above, who spend more (>TWD 10,000), and who are willing to pay for the ecosystem conservation trust fund. This LCM result reflects that the tourist group with obvious preferences has obvious preferences for “reducing the daily tourist limit to 600 tourists”, “improving recreation and facility quality”, and “increasing experience activities to three items” and has lower preference for the “ecosystem conservation trust fund” but does not have a significant preference for “implementation of the tour guide system”. Overall, the interviewed tourist groups at the Orchid Island tourism site exhibited differences in preferences based on their socioeconomic backgrounds and tourism behaviors, and they demonstrated heterogeneity in island tourism attribute preferences. The study by Juutinen et al. [26] on Oulanka National Park in Finland also supports this result.

In conclusion, The study show that CEM can be used to construct a multi-attribute utility function for natural resources and the environment to estimate economic values of goods and services. However, many other attributes could be included, such as ecosystem resilience, beach recreation, and landscape diversity. In this way, the preferences of tourists and local residents for environmental attributes could be better understood. Second, this study could consider local residents to generalize the findings in future. A profound understanding of the determinant variables that affect residents’ attitudes toward tourism development could help community developers and practitioners build a suitably considerate and comprehensive program for future tourism development.

## 6. Conclusions

The development of sustainable island tourism requires the integration of recreation, environment, and management information, which is further considered in the decision-making process for the development and management of sustainable tourism operations. This study used CE to construct a random utility model for the Orchid Island tourism site in Taiwan. To do so, it analyzed various factors, like recreation (such as experience activities), the environment (such as quality of recreation and facilities), systems (such as tourist limit system, tour guide system), and economic considerations (such as the ecosystem conservation trust fund), to construct an evaluation model for validation.

This study summarized five operation and management recommendations as references for the management and operator units of Orchid Island and other relevant industries. This included restricting the daily number of visitors to Orchid Island to 600, implementing relevant measures to improve the quality of recreation facilities, implementing a payment system for professional tour guides, and adding more than three experience activities for island tourism (such as ecotourism, tribal ceremonies, and a cave and underground dwelling experience). These factors would not only increase the overall utility for tourists who come to Orchid Island for island tourism but could also gradually implement an operation and management program for Orchid Island tourism. Secondly, the operators of island tourism should provide in-depth guided experience-tourism services for tourists who are high spenders, tourists aged above 30, and female tourists. Local tour guides could be trained to provide professional guided tours for a target market.

Thirdly, if Orchid Island implements a pricing system, the economic benefits from the aforementioned programs could be combined with its corresponding operation and management costs, and improvements to experience service packages and measures could be included. This could be used to plan specific content for the development of tourism in Orchid Island, which could be used as a reference for determining the costs of island tourism packages. Fourth, it is suggested that, to effectively maintain biodiversity and achieve the goal of conservation and sustainable development, there should be continued use of the Taiwanese National Scenic Area Act and other regulations on land planning and use. Additionally, a more explicit conservation program, geared towards Orchid Island’s regional resource characteristics, should be formulated to realize the long-term preservation of the area’s natural environment, flora, fauna, and historical relics. Moreover, the management of Orchid Island should approve the demolition of illegal buildings or facilities and designate personnel to conduct patrols and inspections to maintain strict control of recreational activities within the park and to prevent improper behavior that might damage or contaminate the environment. Lastly, relevant management units and island tour operators should continue to understand the preferences and attitudes of tourists in the future to propose further operation and management strategies that conform to the concept of island tourism and have more specific and feasible market positioning strategies. This will be more beneficial to the sustainable development of island tourism on Orchid Island.

## Figures and Tables

**Figure 1 ijerph-16-00755-f001:**
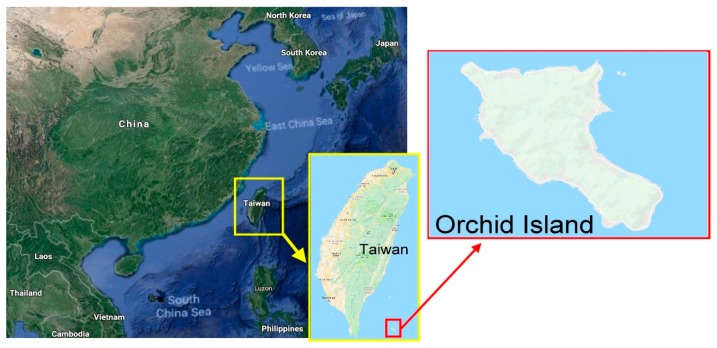
Map of the study area.

**Table 1 ijerph-16-00755-t001:** Attributes and levels of attributes of the Orchid Island tourism site.

Attributes	Levels	Variable	Number of Levels
Limit on the number of visitors	1. Maintaining the status quo: 1000 people per day	LIM^±^	3
	2. 800 people per day (20% reduction)	LIM^−^	
	3. 600 people per day (40% reduction)	LIM^−−^	
Tour guides	1. Maintaining the status quo: professional tour guides not available	GUI^±^	2
	2. Introducing a guided tour facility	GUI^+^	
Recreation and facilities	1. Maintaining the status quo	REC^±^	2
	2. Improving the quality of the recreation and facilities	REC^+^	
Experience activities	1. Maintaining the status quo: snorkeling, whale watching, night observation of flying fish	EXP^±^	4
	2. Addition of any one of the following activities: experiencing ecotourism, tribal ceremonies, or cave and underground dwelling experience	EXP^+^	
	3. Addition of any two of the following three activities: experiencing ecotourism, tribal ceremonies, or cave and underground dwelling experience	EXP^+^	
	4. Addition of the following three activities: experiencing ecotourism, tribal ceremonies and or cave and underground dwelling experience	EXP^+++^	
Ecosystem conservation trust fund	1. Maintaining the status quo: entrance free	FUND	6
	2. TWD 200 per entry per person		
	3. TWD 400 per entry per person		
	4. TWD 600 per entry per person		
	5. TWD 800 per entry per person		
	6. TWD 1000 per entry per person		

**Table 2 ijerph-16-00755-t002:** An example of the choice set of Orchid Island preferences and programs.

Program Attributes	Current Program	Program 1	Program 2	
Limit on the number of visitors	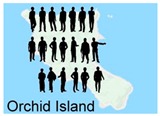 Maintain The current situation	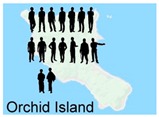 220% reduction	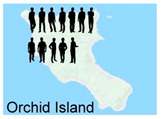 40% reduction	Uncertain
Tour guide	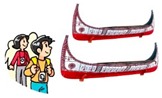 Not available	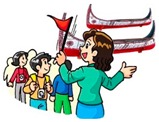 Available	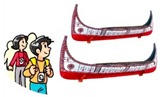 Not available	
Recreation and facilities	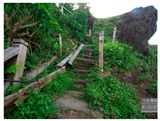 Maintaining the status quo	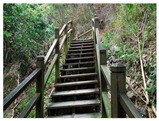 Improved quality	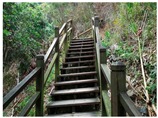 Improved quality	
Experience activities	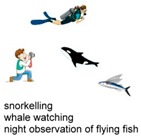 Maintaining the status quo	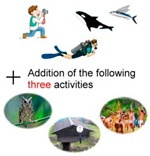 Three additional activities	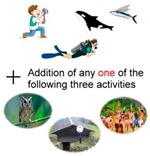 One additional activity	
Ecosystem conservation trust fund (TWD/entrance/person)	Free	TWD 600	TWD 1000	
Please check (1 of 4)	□ Suggestions:	□ Suggestions:	□ Suggestions:	□ Suggestions:

**Table 3 ijerph-16-00755-t003:** Results of the conditional logit model and random parameter logit model.

	Conditional Logit Model		Random Parameter Logit Model			
Attributes and Levels	Coefficient	*t*-Value	Coefficient	*t*-Value	S.E.	*t*-Value
Constant	−0.02	−0.05	−1.34	−3.73 ^c^	3.56	8.46 ^c^
LIM^−−^	0.08	1.12	0.11	0.79	0.49	1.77 ^a^
LIM^−−^	0.15	2.33 ^b^	0.18	1.24	1.24	6.11 ^c^
GUI^+^	0.12	2.71 ^c^	0.12	1.23	0.89	4.66 ^c^
REC^+^	0.28	6.44 ^c^	0.51	5.65 ^c^	0.48	1.60
EXP^+^	−0.02	−0.17	−0.05	−0.30	0.62	1.76 ^a^
EXP^++^	−0.10	−1.31	−0.33	−2.12 ^b^	1.16	4.28 ^c^
EXP^+++^	0.12	1.54	0.26	1.78 ^a^	0.80	2.89 ^c^
FUND	−0.01	−7.21 ^c^	−0.01	−6.69 ^c^		
N of choice sets	1430		1430			
Log-likelihood ratio	−1476.54		−1254.62			
X^2^(0.01,9) = 21.7	614.63 ^c^					

^a^ 10% significance level; ^b^ 5% significance level; ^c^ 1% significance level.

**Table 4 ijerph-16-00755-t004:** Willingness to pay (WTP) for attributes and levels and benefit evaluation of the management program of Orchid Island Tourism.

Attributes and Levels	WTP (TWD/Entrance/Person)	Current Program (TWD/Entrance/Person)	Best Program (TWD/Entrance/Person)	Worst Program (TWD/Entrance/Person)
LIM^±^	−402.72	−402.72		−402.72
LIM^−^	130.26			
LIM^−−^	268.65		268.65	
GUI^±^	−213.62	−213.62		−213.62
GUI^+^	213.62		213.62	
REC^±^	−518.34	−518.34		−518.34
REC^+^	518.34		518.34	
EXP^±^	−0.49	−0.49		
EXP^+^	−19.42			
EXP^++^	−180.32			−180.32
EXP^+++^	201.43		201.43	
Total benefit		−1135.17	1202.04	−1315.21

**Table 5 ijerph-16-00755-t005:** The differences in WTP of Orchid Island visitors from different socioeconomic backgrounds.

	N	Current	LIM^−^	LIM^−−^	GUI^+^	REC^+^	EXP^+^	EXP^++^	EXP^+++^
Men	202	−1226 ^a^	109	190	61 ^b^	541	−46	−392	254
Women	183	−1738	102	191	164	563	−41	−348	263
Age > 30	256	−1602	103	196	162 ^a^	542	−34	−312 ^b^	265
Age ≤ 30	129	−1388	109	187	68	565	−52	−431	254
Tertiary	288	−1486	127 ^a^	186	74	534	−56	−426	226
Secondary & primary	97	−1503	101	195	127	561	−40	−352	271
Income > TWD 30,000	281	−1491	109	164	96	544	−42	−387	261
Income ≤ TWD 30,000	104	−1504	105	211	127	562	−45	−361	262
Cost > TWD 10,000	198	−1786 ^b^	117	224	116	583 ^c^	−28 ^a^	−350	280
Cost ≤TWD 10,000	187	−1210	96	163	116	527	−62	−391	241
WTP Ecosystem conservation trust fund	316	−1929 ^c^	113 ^a^	248 ^c^	129	561	−26 ^c^	−344 ^a^	289 ^c^
Not WTP Ecosystem conservation trust fund	69	138	86	23	72	539	−102	−453	167

^a^ 10% significance level; ^b^ 5% significance level; ^c^ 1% significance level.

**Table 6 ijerph-16-00755-t006:** Evaluation of latent class model (LCM) variables and WTP evaluation of Orchid Island.

Parameters of Attributes and Levels	Coefficient	*t*-Value	WTP
Category 1: Respondents with Strong Preference			
Constant	−0.70	−3.11 ^c^	-
LIM^−^	−0.03	−0.46	-
LIM^−−^	0.26	3.51 ^c^	542.00
GUI^+^	0.04	1.05	-
REC^+^	0.25	5.26 ^c^	532.61
EXP^+^	−0.07	−0.93	-
EXP^++^	−0.15	−1.76 ^a^	−321.36
EXP^+++^	0.18	2.27 ^b^	400.36
FUND	−0.00	−5.69 ^c^	
Category 2: Respondents with a single preference			
Constant	3.58	1.21	-
LIM^−^	0.76	1.24	-
LIM^−−^	−0.93	−1.12	-
GUI^+^	0.80	1.16	-
REC^+^	0.77	1.67 ^a^	395.00
EXP^+^	0.76	0.76	-
EXP^++^	0.94	0.97	-
EXP^+++^	−1.35	−0.83	-
FUND	−0.00	−1.67 ^a^	
Category parameters: Category 1			
Constant	0.41	0.69	
Men	−0.58	−1.82 ^a^	
Age >30	0.16	0.48	
Tertiary	0.65	1.66 ^a^	
Income > TWD 30,000,000	−0.21	−0.65	
Visited Orchid Island before	−0.32	−0.83	
Cost > TWD 2639	0.61	1.76 ^a^	
WTP Ecosystem conservation trust fund	2.54	4.53 ^c^	
N of choice sets		1430	
Log-likelihood ratio		−1563.47	
X^2^(0.01,30) = 50.89		276.00 ^b^	

^a^ 10% significance level; ^b^ 5% significance level; ^c^ 1% significance level.

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
