# Peer review of "Establishment and Application of an Evaluation Model for Orchid Island Sustainable Tourism Development"

_ijerph, 2019, doi:10.3390/ijerph16050755_

Round 1

Reviewer 1 Report

The topic is interesting. It contributes to tourism and hospitality literature in theory and practices. This article is good since it is a useful contribution showing practical application. I can accept this paper being published. The literature review is ok because it supports the theory used for this article. While I support the article, I think language and presentation can be improved by good editing. The “Materials and methods” section is OK! Again, the article would benefit from some editing to make reading easier and shorten the article some. The “Results” section is well presented. The “Discussion” and “Conclusions” sections reflect the research objective in Introduction section. I recommend the paper be accepted for publication.

Author Response

Responses to the comments of Reviewer #1

Thank you very much for your insightful comments and suggestions. We believe
your comments and suggestions are appropriate and useful to us in order to improve
considerably the quality of the manuscript. We have revised our paper in light of your
comments and instructions.

The point-by-point concerns of the reviewers are discussed as follows:
Comments and Suggestions for Authors

Your comments:
The topic is interesting. It contributes to tourism and hospitality literature in theory and
practices. This article is good since it is a useful contribution showing practical
application. I can accept this paper being published. The literature review is ok because
it supports the theory used for this article. While I support the article, I think language
and presentation can be improved by good editing. The “Materials and methods”
section is OK! Again, the article would benefit from some editing to make reading
easier and shorten the article some. The “Results” section is well presented. The
“Discussion” and “Conclusions” sections reflect the research objective in Introduction
section. I recommend the paper be accepted for publication.

Response:
Thank you for your encouragement and comment. The manuscript has been edited by
professional editors to ensure that the language is clear and free of errors, as shown in
comment "CERTIFICATE OF EDITING."

Reviewer 2 Report

A contextual specification based on the different types of evaluation within environmental science would be appropriate (for example: prospective evaluation of alternatives for decision making <ex ante)>

The alternatives for the development of strategies are limited since they focus on market solutions without involving public policy solutions. The above is derived from a need, for the article, to develop more widely a review of Taiwan's public policy regarding its tourism development.

It would be appropriate to point out as a limitation to the study that the opinion of the residents of the island was not available for the creation of sustainable strategies.

In lines 237 and 238 it is not clear whether complete or incomplete questionnaires were excluded.

Reviewer 3 Report

I congratulate the authors for their research. Relevant and novel study (limited to a geographical area), but its conclusions are relevant for destinations with similar characteristics. The authors have used an adequate methodology to comply with the proposed objective. The analysis of the data is adequately explained. The conclusions are relevant and the review of the literature is very extensive.

Next, I propose some improvements:

It is generally recommended to divide the paragraphs. Do not eliminate information, but divide them into separate paragraphs. In the current state many paragraphs are excessively large and make reading difficult. For example, line 59 to 82, is a single paragraph.

Title

It is advisable to add in the title of the article to The islands are known. The article is a proposal of a model to apply in Orchid Island.

Abstract

The abstract is a bit long. It is recommended to shorten: contextualize the topic, objective, methodology and results in a shorter way.

Introduction

The content of the introduction is very complete and adequate. The object of study is well contextualized, what has been done in other studies is exposed. The objective of this investigation in this section. It is very important in this section the objective, not only the summary, the reader needs to know the same. It will include a final paragraph that briefly describes the sections in which the submitted work is divided.

Analysis of the data

The analysis of the data and the methodology is shown as robust from the scientific point of view and is well explained.

Conclusions

The discussion is very complete. The conclusions are relevant and supported in the corresponding results. I advise to improve the conclusions.
